# A chemical screen for modulators of mRNA translation identifies a distinct mechanism of toxicity for sphingosine kinase inhibitors

**Alba Corman**[1], **Dimitris C. Kanellis**[1], **Patrycja Michalska**[1], **Maria Häggblad**[1], **Vanesa Lafarga**[2], **Jiri Bartek**[1,3], **Jordi Carreras-Puigvert**[1¤]*, **Oscar Fernandez-Capetillo**[1,2]*

**1** Science for Life Laboratory, Division of Genome Biology, Department of Medical Biochemistry and Biophysics, Karolinska Institute, Stockholm, Sweden, **2** Genomic Instability Group, Spanish National Cancer Research Centre (CNIO), Madrid, Spain, **3** Danish Cancer Society Research Center, Copenhagen, Denmark

¤ Current address: Department of Pharmaceutical Biosciences and Science for Life Laboratory, Uppsala University, Uppsala, Sweden

* jordi.carreras.puigvert@farmbio.uu.se (JC-P); oscar.fernandez-capetillo@ki.se (OF-C)

**Data Availability Statement:** The authors confirm that all data underlying the findings are fully available without restriction. All relevant data are

## Abstract

We here conducted an image-based chemical screen to evaluate how medically approved drugs, as well as drugs that are currently under development, influence overall translation levels. None of the compounds up-regulated translation, which could be due to the screen being performed in cancer cells grown in full media where translation is already present at very high levels. Regarding translation down-regulators, and consistent with current knowledge, inhibitors of the mechanistic target of rapamycin (mTOR) signaling pathway were the most represented class. In addition, we identified that inhibitors of sphingosine kinases (SPHKs) also reduce mRNA translation levels independently of mTOR. Mechanistically, this is explained by an effect of the compounds on the membranes of the endoplasmic reticulum (ER), which activates the integrated stress response (ISR) and contributes to the toxicity of SPHK inhibitors. Surprisingly, the toxicity and activation of the ISR triggered by 2 independent SPHK inhibitors, SKI-II and ABC294640, the latter in clinical trials, are also observed in cells lacking SPHK1 and SPHK2. In summary, our study provides a useful resource on the effects of medically used drugs on translation, identified compounds capable of reducing translation independently of mTOR and has revealed that the cytotoxic properties of SPHK inhibitors being developed as anticancer agents are independent of SPHKs.

## Introduction

mRNA translation is a fundamental step in gene expression and constitutes the most energy-demanding process in living cells [1,2]. Accordingly, protein synthesis rates are very tightly regulated and rapidly adapt in response to stimuli through the activation of various signaling pathways. For instance, activation of the mechanistic target of rapamycin complex 1 (mTORC1) pathway in response to nutrients or growth factors leads to stimulation of protein synthesis [3]. In contrast, different sources of stress such as amino acid or heme deprivation,

within the paper and its Supporting Information files. Mass Spectrometry data associated to this work are available via ProteomeXchange with identifier PXD017445.

**Funding:** Work related to this work was funded by grants from the Cancerfonden foundation (CAN 2018/381) and the Swedish Research Council (VR) (538-2014-31) to OF. The funders had no role in study design, data collection and analysis, decision to publish, or preparation of the manuscript.

**Competing interests:** The authors have declared that no competing interests exist.

**Abbreviations:** 4E-BP1, 4E-binding protein 1; Act D, Actinomycin D; AML, acute myeloid leukemia; ATF4, activating transcription factor 4; CBCS, Chemical Biology Consortium Sweden; CHX, cycloheximide; DMSO, dimethyl sulfoxide; eIF2α, eukaryotic translation initiation factor 2 subunit 1; ER, endoplasmic reticulum; ERAD, ER-associated protein degradation; FCS, fetal calf serum; GSEA, Gene Set Enrichment Analysis; HPG, homopropargylglycine; HTM, high-throughput microscopy; ISR, integrated stress response; ISRIB, integrated stress response inhibitor; MSigDB, Molecular Signature Database; mTOR, mechanistic target of rapamycin; mTORC1, mechanistic target of rapamycin complex 1; OPP, o-propargyl-puromycin; P70S6K, ribosomal protein S6 kinase β-1; PERK, protein kinase RNA-like endoplasmic reticulum kinase; PERKi, PERK inhibitor; RNP, ribonucleoprotein; RT, room temperature; S1P, sphingosine-1-phospate; SPHK, sphingosine kinase; TEM, transmission electron microscopy; UPR, unfolding protein response; WB, western blotting.

viral infection, and endoplasmic reticulum (ER) stress trigger the so-called integrated stress response (ISR), which, in contrast to mTORC1, lowers global rates of protein synthesis [4]. The relevance for proper translational regulation is evidenced by the fact that alterations in translation have been associated to a wide variety of human diseases, the number of which has recently increased due to advances in genome-wide association studies (reviewed in [5,6]). Cancer, immunodeficiency, and metabolic and neurological disorders are some examples of diseases linked to aberrant protein synthesis. In addition, low levels of translation have been linked to several age-related degenerative conditions.

In what regard to cancer, high cell proliferation rates demand a corresponding increase in the biosynthesis of cellular components, and several observations suggest a contribution of increased translation levels to carcinogenesis. For instance, a number of oncogenes drive transcription of components of the translation machinery and the overexpression of translation initiation factors can facilitate oncogenic transformation (reviewed in [7,8]). In this context, several strategies have focused on lowering translation levels in cancer cells for oncological therapy [9]. For example, chemical or genetic inhibition of the eIF4F complex involved in translational initiation has shown promising results in overcoming the resistance to various cancer therapies, and several compounds are in clinical trials as antineoplastic drugs [10]. An indirect strategy to lower translation is based on small molecules that inhibit rRNA transcription, and compounds such as Actinomycin D (Act D) are currently approved for their use in cancer chemotherapy. Besides drugs that target translation, most therapeutic efforts rely on targeting the signaling pathways that regulate protein metabolism. The best example is that of compounds targeting mTORC1, which are already approved for medical use in oncology and to reduce host versus graft rejection in organ transplants. Another alternative to lower translation rates in cancer cells is through the use of chemicals that activate the ISR [4]. For instance, the FDA-approved drug disulfiram was recently shown to trigger its anticancer effects by triggering an unfolding protein response (UPR) that activates the ISR [11]. In contrast, inactivating the ISR with chemical inhibitors of the protein kinase RNA-like endoplasmic reticulum kinase (PERK) such as integrated stress response inhibitor (ISRIB) has shown particularly promising results in the context of neurodegenerative diseases [6,12,13].

Until recently, measuring overall levels of translation relied on technologies such as polysome profilin or analyzing the incorporation of radioactive amino acids, which have limited throughput. In fact, only 2 chemical screening campaigns directed toward the identification of modifiers of translation have been ever reported [14,15]. However, both of them were done using cell extracts and based on measuring the translation of a specific reporter, and no screen has been yet conducted based on the analysis of overall translation levels in living cells. The recent development of noncanonical amino acids such as puromycin derivatives that can be modified by fluorophores or tags to allow the detection of newly synthesized proteins has now revolutionized the field [16]. These techniques have improved translation analyses by methods such as mass spectrometry, ribosome sequencing, or immunofluorescence [17]. Here, we have capitalized on this technology and conducted a chemical screen to evaluate how 1,200 medically approved compounds, as well as around 3,000 compounds at an advanced level of development, modulate overall levels of translation in mammalian cells.

## Results

### A chemical screen for modulators of protein synthesis

In order to monitor changes in global protein synthesis, we used a labeling method based on o-propargyl-puromycin (OPP) [16]. OPP is an analog of puromycin that is incorporated into the carboxyl terminus of newly synthesized peptides and which can be conjugated to

fluorophores by click chemistry, enabling its visualization by microscopy. First, to determine the suitability of this assay for a chemical screen, we evaluated OPP incorporation in 384-well plates using human osteosarcoma-derived U2OS cells. Image analysis using high-throughput microscopy (HTM) revealed a generalized incorporation of OPP in control wells, which was reduced by treatment with the mTOR inhibitor Torin 2 (0.5 μM, 3 h), and virtually absent with the translation elongation inhibitor cycloheximide (CHX; 100 μg/ml, 30 min) (Fig 1A). To quantify translation levels, we first developed a pipeline that uses the cell nucleus as a starting point from which it expands to the cytoplasm based on the intensity of OPP staining. The resulting region of interest includes both cytoplasmic and nuclear OPP signals. Next, we subtracted the OPP signal from the area occupied by the cell nucleus from the total OPP signal so that the cytoplasmic signal could be measured (Fig 1B). Quantification of HTM images showed a significant reduction in OPP intensity after exposure to controls (Torin 2 and CHX) for both OPP total and cytoplasmic signals (Fig 1C). Nevertheless, since the OPP fraction detected in the nucleus has been previously reported as a result from abortive translation [18] and mRNA translation takes places mainly in the cytoplasm, we used the cytoplasmic OPP signal for subsequent analyses.

Using this setup, we next performed a chemical screen to evaluate how 4,166 pharmacologically active annotated compounds, including approximately 1,200 that are medically approved, modulate mRNA translation in U2OS cells. The screen was conducted in triplicate plates, in which each compound was present at 10 μM for 24 h (Fig 1D). After HTM-mediated quantification, we represented the distribution of the effects of all compounds on the OPP mean intensity (Fig 1E; S1 Table). As expected, the controls Torin 2 and CHX were consistently distributed among the compounds that reduced the OPP signal, with no compound lowering it beyond the effect observed with CHX. For validation experiments, we selected as hits compounds that met 4 criteria: (1) an increase or decrease in OPP intensity greater than 3 standard deviations over the average signal of DMSO controls; (2) an effect in cellular viability not greater than 30% to minimize the impact of compounds lowering translation indirectly through toxic effect of the drugs; (3) that the compound appeared as a hit in all 3 triplicate plates; and (4) that the variation coefficient between the triplicate plates was lower than 20% for both OPP intensity and nuclear counts (viability). With these constraints, 54 compounds that increased OPP incorporation (up-regulators) and 48 compounds that decreased it (down-regulators) were selected as hits for validation experiments (S2 Table). Noteworthy, the effects in lowering translation were overall more profound than those observed on increasing OPP incorporation. In addition, while most down-regulators were annotated as mTOR/PI3K/MAPK inhibitors, there was no enrichment for a specific class of drugs among the potential up-regulators.

## Validation screens failed to identify chemicals increasing translation

Next, we conducted a secondary validation screen with the selected hits, although we reduced the representation of mTOR/PI3K/MAPK inhibitors as the effect of this class of compounds in translation is well established. For validation, each compound was tested in 2 independent plates at 3 concentrations (1, 3, and 10 μM) for 24 h, and the effects in translation were evaluated in 2 orthogonal assays: (1) quantifying OPP incorporation; and (2) measuring the incorporation of homopropargylglycine (HPG), an analog of methionine that, in contrast to OPP, does not arrest translation after its incorporation into elongating polypeptides [19]. While these experiments revealed mild effects for 7 of the up-regulators (S1A Fig; S3 Table), subsequent experiments presented significant variability and inconsistent effects of these compounds in stimulating OPP incorporation. Moreover, none of the compounds affected OPP

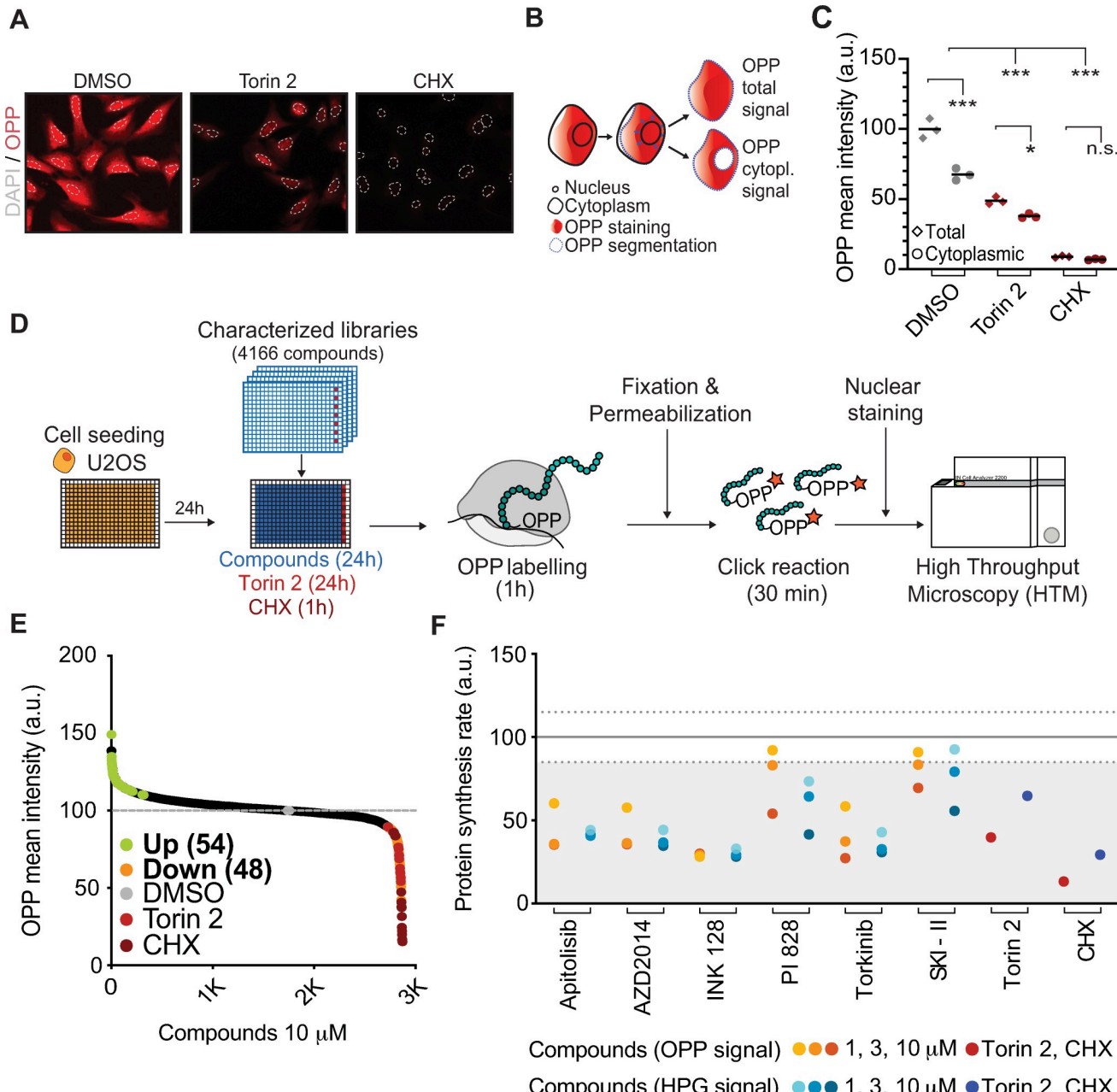

**Fig 1. A chemical screen for modulators of protein synthesis.** (A) OPP incorporation (RED) in U2OS cells exposed to the mTORC1 inhibitor Torin 2 (0.5 μM, 3 h) or CHX (100 μg/mL, 30 min). Nuclei were stained with Hoechst (outlined with a dashed GRAY line). (B) Scheme illustrating the strategy for the quantification of the cytoplasmic OPP signal. (C) HTM-mediated quantification of OPP-associated total and cytoplasmic signal in U2OS cells exposed to Torin 2 and CHX, as in (A). ***$p < 0.001$. Data C in S1 Data. (D) Schematic overview of the phenotypic screen workflow. U2OS cells were plated in 384-well plates. On the next day, cells were exposed to compounds at 10 μM or 0.5 μM of Torin 2 in specific wells (RED) as a control. After 23 h, CHX (100 μg/mL) was added for an hour in specific wells as an additional control (DARK RED). Then, all wells were pulsed with OPP for an hour, after which cells were fixed and processed for HTM-dependent quantification of the OPP signal and nuclei counts. (E) Compound distribution from the screen described above. Compounds increasing (up-regulators, GREEN) or decreasing (down-regulators, ORANGE) OPP incorporation over 3 standard deviations compared to the DMSO control (GRAY) are shown. Compounds exceeding 30% toxicity were excluded from this analysis. Data E in S1 Data. (F) Dose response (1, 3, and 10 μM) of 6 compounds down-regulating translation in U2OS cells after a 24-h exposure. Controls Torin 2 (0.5 μM, 24 h) and CHX (100 μg/mL, 1 h) are shown. Hit validation was conducted by measuring OPP (ORANGE) or HPG (BLUE) incorporation assays. Data F in S1 Data. CHX, cycloheximide; DMSO, dimethyl sulfoxide; HPG, homopropargylglycine; HTM, high-throughput microscopy; mTORC1, mechanistic target of rapamycin complex 1; OPP, o-propargyl-puromycin.

incorporation rates in a shorter treatment (3 h), suggesting that, if any, the effects of these compound in translation after a 24-h treatment would be indirect (S1B Fig).

Since any increase in translation that we ever detected while conducting these experiments was small, we decided to evaluate the feasibility of our image-based approach to detect compounds stimulating translation. To do so, we exposed a panel of different cancer and noncancer cell lines (U2OS, MCF7, and RPE cells) to insulin, which is broadly used as stimulator of mTORC1 signaling, particularly in cells that have been previously starved from nutrients [20]. Activation of mTORC1 in these conditions was confirmed by an increase in the phosphorylation of the ribosomal protein S6 kinase β-1 (P70S6K) and the eukaryotic translation initiation factor 4E-binding protein 1 (4E-BP1), which was prominent in starved cells exposed to insulin but much milder when cells were grown in complete media with 10% of fetal calf serum (FCS) (S1C Fig). As expected, starvation reduced overall translation rates in all cell lines. Interestingly, while insulin increased translation rates in all cells under starvation conditions, it could only do so to a small extent in MCF7 cells grown in 10% FCS and no effect was seen in RPE or U2OS cells. The fact that we conducted our screen in U2OS cells grown in 10% FCS could thus have limited its potential to discover the impact of drugs in up-regulating translation and made us focus on the translation down-regulators.

## The sphingosine kinase inhibitor SKI-II lowers overall translation levels

In contrast to the inconsistent effects observed for translation up-regulators, the dose–response validation experiments mentioned above identified 6 compounds that consistently decreased translation rates in U2OS cells in a dose-dependent manner as measured by either OPP or HPG incorporation (Fig 1E). With the exception of SKI-II, which is an inhibitor of sphingosine kinases (SPHKs) [21], all validated compounds were annotated as mTOR/PI3K/MAPK inhibitors. We thus focused on further characterizing the effects of SKI-II in translation. First, and in order to test whether the effects of SKI-II were direct or indirect, we exposed cells to the compound for 3 h or 24 h. SKI-II significantly decreased OPP incorporation at both time points, although to a lesser extent than Torin 2 (Fig 2A and 2B). Next, we evaluated the impact of SKI-II treatment in the activity of the 2 main signaling cascades controlling translation, mTORC1 and the ISR. Unlike Torin 2, treatment with SKI-II did not affect the phosphorylation of mTORC1 targets such as P70S6K or 4E-BP1. Moreover, SKI-II and Torin 2 had additive effects in lowering translation, further supporting that they modulate translation through independent pathways (S2A Fig). In this regard, and similar to what is observed with Tunicamycin, a well-established activator of the ISR, SKI-II triggered phosphorylation of the eukaryotic translation initiation factor 2 subunit 1 (eIF2α) and increased the levels of activating transcription factor 4 (ATF4) (Fig 2C). Moreover, SKI-II treatment led to the nuclear accumulation of ATF4, which occurs subsequent to eIF2α phosphorylation during activation of the ISR (Fig 2D).

Next, we explored whether ISRIBs could revert the effects of SKI-II in translation. In addition to increasing eIF2α phosphorylation and ATF4 levels, exposing U2OS cells to SKI-II or Tunicamycin induced phosphorylation of PERK kinase, a critical mediator of the ISR, as evidenced by its reduced mobility in western blotting (WB) (Fig 2E). Moreover, exposure to the PERK inhibitor (PERKi) GSK2606414 [22] blocked PERK activation, phosphorylation of eIF2α, and the increase in the levels of ATF4 induced by SKI-II. In contrast, ISRIB, which inhibits the ISR downstream of PERK and eIF2α [23], solely prevented the accumulation of ATF4 but not PERK or eIF2α phosphorylation. Consistent with WB data, HTM-mediated analysis showed that PERKi and ISRIB prevented the nuclear translocation of ATF4 induced by SKI-II (Fig 2F). Importantly, PERKi and ISRIB reverted the down-regulation of protein

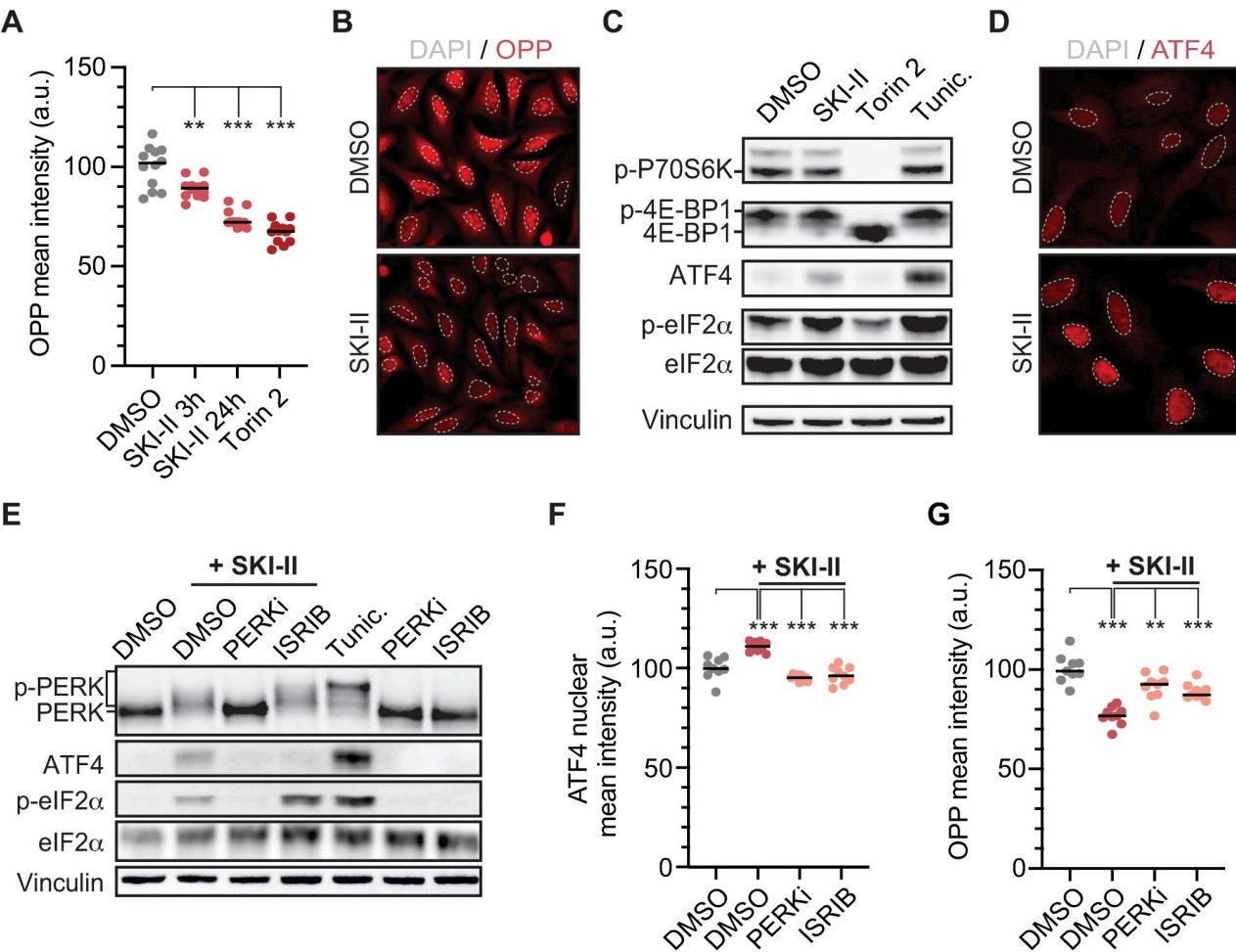

**Fig 2. SKI-II reduces mRNA translation by activating the ISR.** (A) HTM-mediated quantification of the mean intensity of OPP signal in U2OS cells exposed to SKI-II (10 μM) at different time points (3 and 24 h). Torin 2 (0.5 μM, 3 h) was included as a control. Data A in S2 Data. (B) Reduction in OPP signal (RED) in U2OS cells exposed to SKI-II (10 μM) for 3 h compared to the DMSO control. Nuclei were stained with Hoechst (dashed GRAY line). (C) WB for markers of the mTORC1 (p-P70S6K and 4E-BP1) or ISR (p-eIF2α and ATF4) pathways in U2OS cells exposed during 3 h to SKI-II (10 μM), Torin 2 (0.5 μM). The ISR-activating compound Tunic. (10 μg/mL) or the mTORC1 inhibitor Torin 2 were used as controls (0.5 μM, 3 h). Total levels of Vinculin and eIF2α are shown for normalization. (D) Nuclear accumulation of ATF4 in U2OS cells exposed to SKI-II (10 μM) for 3 h. (E) WB of markers of activation of the ISR (p-PERK, ATF4, and p-eIF2α) in U2OS cells preexposed for 1 h to PERK inhibitor (PERKi, 0.5 μM) and ISRIB (50 nM) or DMSO, and then exposed to SKI-II (10 μM) or DMSO for 3 h. The p-PERK is recognized as a shift in the migration pattern of the protein. (F) HTM-mediated quantification of ATF4 mean nuclear intensity in U2OS cells treated as in (E). Data F in S2 Data. (G) HTM-mediated quantification of OPP signal in U2OS cells treated as in (E). For the HTM-mediated quantifications, every dot represents the mean value of the measurements taken per well from 3 independent experiments. Data G in S2 Data. Statistical analyses were done using one-way ANOVA tests. ***$p < 0.001$. 4E-BP1, 4E-binding protein 1; ATF4, activating transcription factor 4; DMSO, dimethyl sulfoxide; eIF2α, eukaryotic translation initiation factor 2 subunit 1; HTM, high-throughput microscopy; ISR, integrated stress response; ISRIB, integrated stress response inhibitor; mTORC1, mechanistic target of rapamycin complex 1; OPP, o-propargyl-puromycin; p-eIF2α, phosphorylated eIF2α; p-P70S6K, phosphorylated P70S6K; PERK, protein kinase RNA-like endoplasmic reticulum kinase; PERKi, PERK inhibitor; p-PERK, phosphorylated PERK; Tunic., Tunicamycin; WB, western blotting.

synthesis induced by SKI-II, as measured by OPP incorporation (Fig 2G). To evaluate the effects of SKI-II on translation in an independent assay, we performed polysome analyses in the breast cancer cell line MCF7. Of note, we chose MCF7 as these cells were more suited for polysome profiling than U2OS, and OPP incorporation experiments confirmed a similar effect of SKI-II in reducing translation in both cell types (S2B Fig). Consistent with OPP data, exposure to SKI-II resulted in an accumulation of 80S monosomes in MCF7 cells that was

associated to a loss of polysomes, which was reverted by a concomitant treatment with PERKi (S2C Fig). Collectively, these data identify SKI-II as a chemical that lowers overall translation levels in human cells, which is mediated by the activation of the ISR.

## SKI-II damages the membranes of the endoplasmic reticulum (ER)

To better understand the mechanism by which SKI-II impacts on translation, we conducted proteomic analysis in U2OS cells exposed to this compound (10 μM, 6 h). SKI-II significantly modified the levels of 43 proteins. Interestingly, some of the most highly up-regulated proteins such as HERPUD, CRE3BL2, XBP1, and DNAJB9 are associated to pathways related to the ISR-activating ER stress response, including the UPR and ER-associated protein degradation (ERAD) system (Fig 3A). Accordingly, Gene Set Enrichment Analysis (GSEA) revealed a significant enrichment of "ER-unfolded protein response" and "negative regulation of response to ER stress" pathways in response to SKI-II (Fig 3B). Thus, the main cellular effects of the SPHK inhibitor SKI-II are related to a specific perturbation of the ER, which leads to the activation of ER stress responses. Supporting this view, transmission electron microscopy (TEM) of U2OS cells treated with SKI-II revealed an enlargement of the ER cisterns already evident after 3 h, which was exacerbated at times 6 h and 24 h (Fig 3C). This phenomenon was accompanied by a loss of ribosomes from ER membranes, likely contributing to the overall reduction in protein synthesis triggered by SKI-II.

Next, given that SKI-II is a SPHK inhibitor and that sphingolipids and their derivative metabolites form part of cellular membranes [24], we evaluated if the effect of SKI-II on the ER was also observed in other membrane-associated organelles. To do so, we used Cell Painting, a technique developed to label cellular organelles for their analysis by fluorescence microscopy [25]. HTM-mediated quantification of Cell Painting in U2OS cells treated with SKI-II revealed a significant loss of the ER signal (Figs 3D and S3A), with no apparent changes in the mean intensity of the signal associated to the Golgi apparatus or mitochondria (S3B and S3C Fig). Of note, changes in ER morphology can be produced directly by changes in the cellular lipid composition or as a consequence of an activation of PERK-dependent signaling of the UPR [26,27]. In this regard, the reduction in the ER-associated signal in cells treated with SKI-II was not rescued by the inhibition of PERK, arguing that this is a primary effect of the compound on ER membranes that would then trigger the activation of the ISR.

To further evaluate the effects of SKI-II on ER membranes and the ISR, we compared the effects of SKI-II to those of 2 known activators of the UPR, Thapsigargin, a compound that inactivates $Ca^{2+}$-dependent chaperones necessary for protein folding in the ER [28], and Tunicamycin, an inhibitor of N-glycosylation, which is necessary for correct protein folding [29]. While all compounds reduced the ER signal as measured by Cell Painting to a similar extent, PERK inhibition only rescued the ER signal after treatment with Thapsigargin and Tunicamycin, having no effect in the context of SKI-II (Fig 3E). TEM images of cells exposed to Tunicamycin confirmed the effects of the compound on the ER (S3D Fig) and in inhibition of global protein synthesis through activation of the ISR, which was prevented by PERKi and ISRIB (S3E and S3F Fig). Together, these results support that the primary effect of SKI-II is a specific damage to ER membranes, which triggers the activation of the ISR thereby reducing translation rates in mammalian cells.

## SKI-II and its clinical derivative ABC294640 activate the ISR and kill cells in the absence of SPHKs

Besides reverting its effects on translation, we noted that inhibition of the ISR with either PERKi or ISRIB also partially rescued the toxicity of Tunicamycin (S3G and S3H Fig). Given

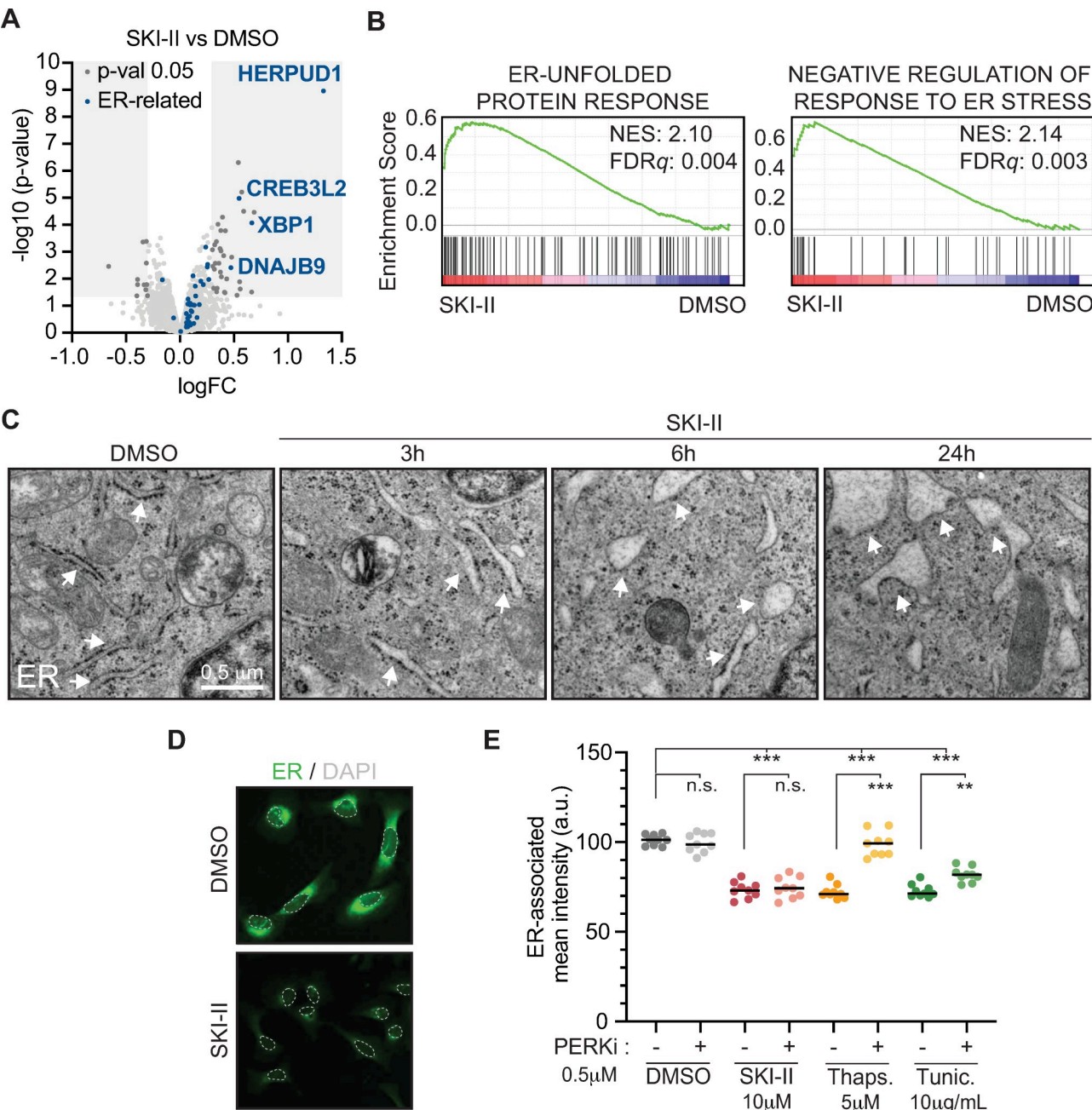

**Fig 3. SKI-II specifically affects ER membranes.** (A) Volcano plot representing the data from a proteomic experiment in U2OS cells treated with SKI-II (10 μM, 6 h) compared to a DMSO-treated control. Proteins within the GRAY boxes are differentially regulated ($p < 0.05$). Factors in BLUE are related to the ER stress response. Data A in S3 Data. (B) GSEA from the experiment defined in (A) showing a significant enrichment in GO terms "ER-unfolded protein response" (NES: 2.10, FDRq: 0.004) and "negative regulation of response to ER stress" (NES: 2.14, FDRq: 0.003) for cells exposed to SKI-II. Data B in S3 Data. (C) TEM images of U2OS cells treated with SKI-II (10 μM) for 3, 6, and 24 h. Cells were exposed to DMSO as a control. White arrows point at the ER. Ribosomes are black electrodense particles situated around the cisterns of the ER. Scale bars, 0.5 μm. (D) Representative image illustrating the reduction in ER-associated signal (GREEN) in U2OS cells exposed to SKI-II (10 μM) for 3 h compared to the DMSO control. Nuclei were stained with Hoechst (dashed GRAY line). (E) HTM-dependent quantification of ER-associated mean intensity in U2OS cells exposed to PERKi (0.5 μM) for 1 h and then to SKI-II (10 μM), Thaps. (5 μM), or Tunic. (10 μg/mL) for additional 3 h. In HTM quantifications, every dot represents the mean value of the measurements taken by well from 3 independent experiments. Data E in S3 Data. Statistical analyses were done using one-way ANOVA tests. ***$p < 0.001$. DMSO, dimethyl sulfoxide; ER, endoplasmic reticulum; GO, Gene Ontology; GSEA, Gene Set Enrichment Analysis; HTM, high-throughput microscopy; n.s., not significant; PERKi, PERK inhibitor; TEM, transmission electron microscopy; Thaps., Thapsigargin; Tunic., Tunicamycin.

the similarities we found between Tunicamycin and SKI-II, we revisited the mechanism of toxicity that is currently accepted for SPHK inhibitors, which relates to the accumulation of ceramides as inductors of apoptosis [30]. Supporting that activation of the ISR contributes to the toxicity of SKI-II, nuclear counting experiments revealed that PERKi or ISRIB reduced the toxicity of SKI-II in U2OS cells (Fig 4A). Similarly, PERKi or ISRIB limited the toxicity of SKI-II in a mouse cell line of acute myeloid leukemia (AML), which is relevant given that SKI-II was shown to be particularly efficacious for killing AML cells in vitro and in vivo [31] (Fig 4B).

SKI-II is a first-generation SPHK inhibitor, an analog of sphingosine that inhibits both SPHK1 and SPHK2, with more preference for the latter in biochemical assays [21,32]. Furthermore, SKI-II has been shown to inhibit desaturase 1 (DEGS1), which is also involved in the synthesis of sphingolipids [33]. We thus wanted to clarify which of these targets was responsible for the effects of SKI-II in toxicity and translation. To do so, we exposed U2OS cells to SKI-II or to selective inhibitors of SPHK1 (PF-543) [34], SPHK2 (ABC294640) [35], and DEGS1 (DESi) [33] for 3 h. Interestingly, only the treatment with ABC294640 led to a significant increase in nuclear ATF4 (Fig 4C) and a reduction in ER signal (Fig 4D), suggesting that SPHK2 could be the relevant target of these compounds on activating the ISR. Importantly, ABC294640 is a structural analog of SKI-II, which was developed to be more selective for SPHK2 than for SPHK1, and is currently in clinical trials for cancer chemotherapy [33]. Based on our previous results with SKI-II, and given the clinical relevance of ABC294640, we set to explore whether its toxicity was also related to an activation of the ISR. Supporting this view, exposure to ABC294640 led to lower OPP incorporation rates (S4A Fig), and a concomitant treatment with PERKi or ISRIB rescued the nuclear accumulation of ATF4 triggered by ABC294640 (S4B Fig).

Finally, and to further investigate the role of each SPHK on the cellular effects of SKI-II and ABC294640, we generated U2OS cells lacking SPHK1, SPHK2, or both by CRISPR-mediated gene editing. Surprisingly, the deficiency of even both SPHKs did not lead to an activated ISR as measured by ATF4 levels of eIF2α phosphorylation (Figs 4E and S4C). Moreover, the lack of SPHK1 and SPHK2 did not prevent the effects SKI-II and ABC294640 on activating the ISR (Figs 4E and S4C), on the ER (Fig 4F), and on lowering translation rates (Figs 4G and S4D). What is most important, the toxic effects of both drugs were observed regardless of both SPHK1 and SPHK2 (Figs 4H and S4E). Collectively, these experiments indicate that the toxicity of SPHK inhibitors SKI-II and ABC294640 is due to an effect of these compounds on ER membranes that activates the ISR and that it is independent of SPHKs.

## Discussion

We here present a resource that illustrates the effects of over 4,100 compounds, including approximately 1,200 medically approved drugs, on overall levels of translation. It is important to note that this is the first chemical screen that has measured total translation levels in living cells, instead of relying on in vitro assays or measuring the translation of a specific reporter gene. Our screen failed to identify compounds capable of consistently stimulating translation. In fact, while translation inhibitors have been broadly developed [9], there are very few examples described to enhance protein production, always at modest levels, and mostly in conditions where translation levels were previously reduced. For instance, while overexpression of the translation factor eIF1A rescues translation in a cellular model of amyotrophic lateral sclerosis where protein synthesis was compromised, it fails to stimulate translation in control conditions [36]. In addition, whereas inhibition of the ISR kinase PKR by 2 small molecules was found to stimulate translation in rabbit reticulocyte extracts, this was due to a basal activation

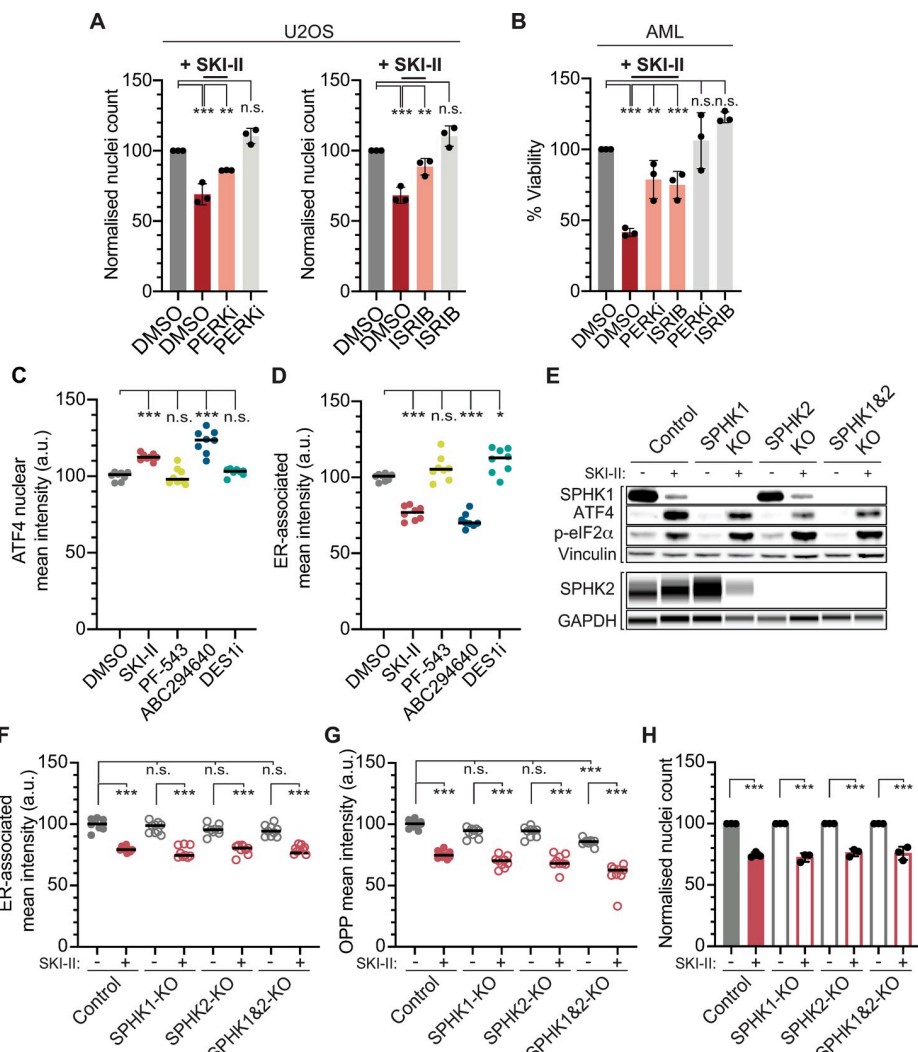

**Fig 4. SKI-II lowers translation and kills cells independently of SPHKs.** (A) HTM-dependent quantification of nuclei numbers (normalized to DMSO) in U2OS cells exposed to SKI-II (10 μM), alone or in combination with PERKi (0.5 μM) or ISRIB (10 nM) for 24 h. Cells were also solely exposed to PERKi and ISRIB at the indicated concentrations. Data A in S4 Data. Statistical analyses were done using one-way ANOVA tests. $^{***}p < 0.001$. (B) Percentage of cellular viability (normalized to DMSO) measured with CTG in mouse AML cells exposed to SKI-II (1 μM), alone or in combination with PERKi (3 μM) or ISRIB (10 nM) for 24 h. Treatment with solely PERKi and ISRIB at indicated concentrations is shown. Data B in S4 Data. Statistical analyses were done using one-way ANOVA tests. $^{***}p < 0.001$. (C) HTM-mediated quantification of ATF4 nuclear signal in U2OS cells treated with SKI-II (10 μM), the SPHK1 inhibitor PF-543 (10 μM), the SPHK2 inhibitor ABC294640 (50 μM), or the DES1 inhibitor DESi (10 μM) for 3 h. Data C in S4 Data. (D) HTM-mediated quantification of the mean intensity of the ER signal (measured by Cell Painting) in U2OS cells treated as in (D). Data D in S4 Data. (E) WB of markers of activation of the ISR (ATF4 and p-eIF2α) in parental U2OS cells (control) and in U2OS cells deficient for SPHK1 (SPHK1 KO), SPHK2 (SPHK2 KO), or both (SPHK1&2 KO), as shown by SPHK1 and SPHK2 antibodies, exposed to SKI-II (10 μM, 3 h) for 3 h. (F) HTM-dependent quantification of ER-associated mean intensity in parental U2OS cells (control) and in U2OS cells deficient for SPHK1 (SPHK1 KO), SPHK2 (SPHK2 KO), or both (SPHK1&2 KO) exposed to SKI-II (10 μM, 3 h). Data F in S4 Data. (G) HTM-mediated quantification of OPP signal in cells treated as in (G). In HTM quantifications, every dot represents the mean value of the measurements taken by well from 3 independent experiments. Data G in S4 Data. Statistical analyses were done using one-way ANOVA tests. $^{***}p < 0.001$.(H) HTM-dependent quantification of nuclei numbers (normalized to control) of parental U2OS cells (control) and in U2OS cells deficient for SPHK1 (SPHK1 KO), SPHK2 (SPHK2 KO), or both (SPHK1&2 KO) exposed to SKI-II (10 μM) for 24 h. Data H in S4 Data. Statistical analyses were done using one-way ANOVA tests. $^{***}p < 0.001$. AML, acute myeloid leukemia; ATF4, activating transcription factor 4; CTG, Cell Titer Glo; DMSO, dimethyl sulfoxide; ER, endoplasmic reticulum; GAPDH, glyceraldehyde 3-phosphate dehydrogenase; HTM, high-throughput microscopy; ISR, integrated stress response; ISRIB, integrated stress response inhibitor; KO, knockout; n.s., not significant; OPP, o-propargyl-puromycin; p-eIF2α, phosphorylated eIF2α; PERKi, PERK inhibitor; SPHK, sphingosine kinase; WB, western blotting.

of the ISR in the in vitro system and was not reproduced in unchallenged cells [15]. Consistently, our work and others show that inhibition of the ISR, using PERKi or ISRIB, does not increase protein synthesis in the absence of stress [23]. Arguably, the best-known example of a molecule capable of stimulating translation is insulin. However, the effects of insulin in inducing translation are observed primarily in cells undergoing starvation, which we could confirm in this study [37]. All things considered, our results suggest that substantially up-regulating translation might be challenging to achieve with chemicals, particularly in cancer cells grown in full media, as translation utilizes the largest fraction of the energy in living cells [1]. An independent screen in starved primary cells could nevertheless be of interest to discover compounds that can stimulate translation in stressed conditions.

As for molecules reducing translation rates, the fact that most compounds that significantly lowered translation were annotated as mTOR/PI3K/MAPK inhibitors reinforces the central role of this pathway in protein synthesis. In addition, our screen identified the SPHK inhibitor SKI-II as a novel inhibitor of protein synthesis. Our data are consistent with previous observations of ER stress in response to SKI-II, particularly in combination with temozolomide or bortezomib [38,39]. Our study is nevertheless first to report the effects of SKI-II and its clinical derivative ABC294640 in translation and has also helped to clarify mechanism of action of SPHK inhibitors. The interest in SPHKs as anticancer targets started with the observation that these enzymes are frequently overexpressed in tumors (reviewed in [30]). The original and still widely accepted hypothesis was that inhibition of SPHKs would result in lower levels of the pro-survival factor sphingosine-1-phospate (S1P) while increasing the levels of the apoptotic factor ceramide. Based on this line of thought, several inhibitors of SPHK1 and SPHK2 have been discovered and are at various stages of development as clinical drugs. However, the role of the S1P/ceramide balance and the contribution of each SPHK to the toxicity of these compounds remained to be fully confirmed.

The initial compounds targeting SPHKs, such as SKI-II, were inactive sphingosine analogs that targeted both SPHK1 and SPHK2. However, while early studies pointed to SPHK1 as the relevant anticancer target, later works casted doubts on such a view [40,41]. For instance, even if SPHK1 or SPHK2 depletion both cause defects in cell growth, these are more prominent when targeting SPHK2, yet only SPHK1 depletion leads to a significant increase in ceramides [42–45]. Moreover, and contrary to depletion experiments, use of the SPHK1 selective inhibitor PF-543 fails to induce cell death in different cancer cell lines [46]. These results suggested that perhaps SPHK2 was the main target responsible for the toxicity of SPHK inhibitors and further raised doubts on whether the S1P/ceramide balance is the main determinant of their toxicity. Our study puts an end to this discussion as we unambiguously demonstrate that SKI-II and ABC294640 kill SPHK1- and/or SPHK2-deficient cells as effectively as wild-type cells. This knowledge is particularly relevant for ABC294640, as it is currently being tested in several clinical trials for cancer and also for the treatment of COVID-19-associated pneumonia [47]. In any case, a better understanding of the mechanism of action of drugs is always be beneficial to improve their potential uses. For instance, we now know that the toxicity of these compounds (and that of Tunicamycin) is alleviated by a concomitant treatment with PERKi or ISRIB, although whether this is related to translation remains unknown. In addition, these results provide a rationale for drug combinations with SKI-II and ABC294640, the effects of which should be particularly exacerbated by other chemicals that activate ER- or unfolded proteins responses. On the other hand, and despite ISR activators might be toxic for cancer cells, we should note that there are also disorders where an activation of the ISR has been proposed to be beneficial, such as Charcot–Marie–Tooth disease, opening new areas where these compounds could be of interest [12,48]. To end, compounds such as mTOR inhibitors are widely used in biomedical research to revert age-related pathologies, and some have obtained clinical

approval in the context of oncology and organ transplant. To what extent other drugs that lower overall translation levels independently of mTORC1 might be of use in these or other pathologies emerges as an interesting possibility.

## Material and methods

### Cell lines

Human female osteosarcoma U2OS and epithelial RPE-1 cells were obtained from ATCC, the breast cancer MCF7 (kind gift of Dr Thomas Helleday, Karolinska Institutet, Stockholm, Sweden) cells were cultured in DMEM + Glutamax (Thermo Fisher Scientific, 31966–047) supplemented with 10% FCS and 1% Penicillin/Streptomycin. Mouse AML cells [49] were cultured in RPMI1640 (Thermo Fisher Scientific, 3196621875034) and also supplemented with serum and antibiotics. For the validation screen, DMEM lacking methionine (Thermo Fisher Scientific, 21013024) and supplemented with 2 mM L-glutamine (Sigma-Aldrich, G7513) and 5 mM L-Cysteine (Sigma-Aldrich, C6727) was used.

### Generation of SPHKs knock-out cell lines

For knocking out SPHK1 and SPHK2, we used sgRNA sets designed by SYNTHEGO (3 sgRNA per target) and an additional set of control nontargeting sgRNAs (TRAC). The following sgRNA sequences were used: SPHK1 5′-UUUUCUCAGCGGGCGGCCCC-3′, 5′-GCAA GGCCUUGCAGCUCUUC-3′ and 5′-ACCAGUGAGCAUCAGCGUGA-3′; SPHK2 5′-CGC AGGCCCUGCACAUACAG-3′, 5′-UGGCCUGGUCCCGUUGGCCG-3′ and 5′-CCGCUG AGUCUGAGGGGCUG-3′. Briefly, sgRNAs were coupled with Cas9 2NLS nuclease (*S. pyogenes*) (SYNTHEGO) forming ribonucleoprotein (RNP) complexes, and cells were transfected using Lipofectamine CRISPRMAX Cas9 Transfection Reagent (Thermo Fisher, Sweden). U2OS cells were seeded on top of the transfection mixes (30,000 cells/well) in 12-well plates. Single-cell clones were generated and screened for finding single knock-outs of SPHK1 and SPHK2. For producing the double knock-out, SPHK1-KO selected clones were transfected with sgRNAs targeting SPHK2 as already stated and vice versa for SPHK2-KO clones. Then, single clones were generated and screened. The same procedure was done using cells transfected with nontargeting sgRNAs as a control.

### Compounds

Torin 2 (SML1224), CHX (C7698), insulin solution (I0516), Clomiphene citrate (C6272), ARC239 dihydrochloride (A5736), Imatinib (SML1027), UNC1999 (SML0778), SKI-II (S5696), and ISRIB (SML0843) were purchased from Sigma-Aldrich (Germany); NVP-BVU972 (S2761) and FK-506 (Tacrolimus, S5003) were obtained from Selleck Chemicals (USA); JNK Inhibitor V (420129) from Calbiochem (USA); PERK inhibitor GSK2606414 (PERKi, 516535) from Merck Millipore; Tunicamycin (11445) from Bionordika (Sweden); Thapsigargin (ab120286) from Abcam; PF-543 hydrochloride (5754) from R&D Systems (United Kingdom); ABC294640 (10587) from Cayman Chemical (USA); and the DES1 inhibitor (B-0027) from Echelon Biosciences (USA).

### Chemical screens

Plate and liquid handling were performed using Echo550 (Labcyte, USA), Janus Automated Workstation (PerkinElmer, USA), Multidrop 384 (Thermo Scientific, Sweden), Viaflo 384 (Integra Biosciences, Japan), Multiflo FX Multi-Mode Dispenser (BioTek, USA), and Hydro-speed washer (Tecan, Switzerland). Cells were seeded in black with clear bottom 384-well

plates (BD Falcon, 353962). Compound libraries were provided by the Chemical Biology Consortium Sweden (CBCS). The chemical collection used in the primary screening contained 4,166 pharmacologically active compounds from the following libraries: Prestwick Chemical Library of FDA-approved compounds, Tocris mini, Selleck tool compounds, Selleck-known kinase inhibitors, ENZO tool compounds, SGC bromodomain probes, and 115 covalent drugs synthesized by M. Henriksson (Karolinska Institutet, Sweden).

For the primary screen, U2OS cells were trypsinized and resuspended in culture medium. The cell suspension (750 cells in 30 μl/well) was dispensed into 384-well plates and incubated overnight at 37˚C in a 5% $CO_2$ atmosphere. The following day, cells were exposed to a final concentration of 10 μM of compounds diluted in dimethyl sulfoxide (DMSO). Compound addition was done in triplicate sets of plates. At the same time, the mTORC1 inhibitor Torin 2 (SML1224, Sigma-Aldrich) was added as a positive control in specific wells at a final concentration of 0.5 μM. After 23 h, CHX (C7698, Sigma-Aldrich) was added as an additional control for 1 h to a final concentration of 100 μg/mL in specific wells. Cells were then pulsed with OPP (Jena Bioscience, Germany) dissolved in media at a final concentration of 20 μM for 1 h at 37˚C. After that, plates were washed once with PBS and cells fixed with 100% ice-cold methanol for 20 min at room temperature (RT). After fixation and washing, cells were permeabilized with Triton X-100 (0.1%) for 20 min, RT. Next, cells were incubated with Click reaction cocktail (88 mM Na-phosphate (pH 7), 20 mM $CuSO_4$, 10 mM Na-Ascorbate, 2 μM Alexa Azide 647) for 30 min in the dark. Nuclei were stained with 2 μM Hoechst 33342 for 15 min in the dark. Plates were imaged using an IN Cell Analyzer 2200 system with a 10× objective, 4 images per well were acquired, covering the whole well. Images were analyzed with CellProfiler using a custom-made pipeline for the detection of cytoplasmic OPP signal. The pipeline uses the nucleus of the cell as a seed, which expands a defined number of pixels to the cytoplasm based on the intensity of OPP signal, generating a secondary object that includes both the nucleus and the cytoplasmic OPP signal (OPP total). To quantify the OPP cytoplasmic signal, a tertiary object (OPP cytoplasm) is created substracting the nucleus from the secondary object (OPP total). Then, the intensity of OPP signal is measured in the object OPP cytoplasm. Adequate controls, Torin 2, and CHX were used for setting up the analysis pipeline. The analysis considered the mean intensity of cytoplasmic OPP staining and the nuclei count. All values were normalized to DMSO conditions within each plate. Then, the mean value for each compound in triplicates was calculated, representing a single measurement per compound per set of triplicates. Images were acquired using an IN Cell Analyzer 2200 (GE Healthcare, USA), and quantitative image analyses were run in the open-source software CellProfiler (www.cellprofiler.org) [50]. Statistical analysis of high content imagining data was conducted using TIBCO Spotfire, and additional statistical analyses were done using Microsoft Excel or Graphpad Prism software.

For the validation screen, U2OS cells were exposed to 3 concentrations of the selected hits (final concentration 1, 3, and 10 μM) for 24 h. Translation rates were evaluated quantifying both OPP and HPG incorporation. The validation with OPP and HPG was conducted in duplicates. Prior to the HPG pulse, cells were washed with PBS and media was exchanged to DMEM lacking methionine for 30 min. HPG diluted in DMEM lacking Met was added to cells to a final concentration of 10 μM. Cells were kept at 37˚C for 30 min. For HPG incorporation measurements, the Click-iT HPG Alexa Fluor 488 Protein Synthesis Assay Kit was used (Thermo Fisher Scientific, C10186).

## OPP labeling

Briefly, U2OS (4,000 cells/well) were plated onto imaging 96-well plates (BD Falcon, 353219). The next day, cells were either preexposed to PERKi or ISRIB for 1 h followed by addition of

inducers of integrated stress (Tunicamycin and Thapsigargin) or SPHK inhibitors (SKI-II, ABC294640) for 3 h, or were exposed to the ISR inducers and SPHKs and DES1 inhibitors (SKI-II, ABC294640, PF-543, and DES1i) alone, at indicated concentrations and times. For experiments involving up to 24-h exposure to compounds, U2OS (2,000 cells/well) and MCF7 (3,500 cells/well) were seeded. For experiments comparing translation rates in cells grown in complete media (10% FCS) or starvation (0% FCS), cells were seeded in complete media: U2OS (2,000/4,000), MCF7 (4,000/8,000), and RPE-1 (4,000/8,000) cells/well; after 24 h, cells were washed in PBS and media was exchange to 10% FCS/0%FCS; cell densities were adjusted to maintain similar cell count under conditions of growth and starvation. After the corresponding treatments, cells were pulsed with OPP for 30 min, fixed with 100% methanol, and click chemistry was conducted following manufacturer's instructions for Click-iT OPP Alexa Fluor 647 Protein Synthesis Assay Kit (Thermo Fisher Scientific, C10458). Nuclei were stained with Hoechst.

## Immunofluorescence

U2OS cells were seeded and treated as above onto 96-well imaging plates. Then, cells were fixed with 4% PFA for 15 min and permeabilized with 0.1% Triton X-100 for 10 min at RT. After blocking (3% BSA and 0.1% Tween-20 in PBS) for 30 min, the indicated concentration of primary antibodies was applied: ATF4 (D4B8) (1:200, Cell Signaling Technology, 11815S). The following secondary antibodies were added for 1 h at RT: anti-rabbit IgG-647 (1:500, Life Technologies, A-21244) and anti-rabbit IgG-488 (1:500, Life Technologies, A-11008). Images were acquired with IN Cell. Nuclear translocation of ATF4 was measured by increase of ATF4 mean intensity into the nucleus of cells using Hoechst as a counterstaining. Cell painting [25] was used to differentially stain membrane-bound organelles. Before fixation, MitoTracker Deep Red A-647 (Thermo Fisher Scientific, M22426) was diluted in DMEM (1:1,000) and added to live cells for 20 min at 37°C. Afterwards, cells were washed with PBS, fixed and permeabilized as before, and stained with a cocktail of Hoechst (1:1,000), Concanavalin A A-488 (1:200, Thermo Fisher Scientific, C11252), and Wheat Germ Agglutinin A-555 (1:500, Thermo Fisher Scientific, W32464) for 20 min in the dark. Image analysis was performed using a self-made pipeline identifying the different organelles according to intensity of the signal in the cytoplasm. In these experiments, 9 pictures were taken per well using a 20× objective.

## Western blotting

For experiments where U2OS cells were exposed to compound for less than 24 h, 60,000 cells/well were seeded onto 12-well plates. For the experiments comparing cells growing in 10% FCS/starvation for all U2OS (30,000/60,000 cells/well), MCF7 and RPE (40,000/80,000 cells/well). RIPA buffer with protease and phosphatase inhibitors (Sigma, Germany) was used for preparing protein lysates from U2OS cells treated as indicated. Immunoblotting was performed following standard protocols with indicated antibodies: ATF4 (D4B8) (1:500 or 1:50, Cell Signaling Technology, 11815S), p-P70S6K (Thr389) (1:1,000, Cell Signaling Technology, 9205), 4E-BP1 (1:1,000, Cell Signaling Tech, 9452S), p-eIF2α (phospho S51) [E90] (1:1,000, Abcam, ab32157), eIF2α (1:500, Cell Signaling Technology, 9722), Vinculin (EPR8185) (1:2,000, Abcam, ab129002), PERK (D11A8) (1:500, Cell Signaling Technology, 5683), SPHK1 (D1H1L) (1:500, Cell Signaling Technology, 12071S), SPHK2 (1:50, Abcam, ab37977), and GAPDH (1:100, Abcam, ab9485). Protein bands were visualized by chemiluminescence (either with ECL, Thermo Scientific, 34076, or Amersham ECL, GE Healthcare, RPN2235) and imaged on an Amersham Imager 600 (GE Healthcare, USA). In order to detect changes in

SPHK2, and ATF4 simultaneously, samples were analyzed using JESS/WES immunoblot system (ProteinSimple, USA).

## Viability experiments

U2OS (2,000 cells/well) and mouse AML cells (10,000 cells/well) were seeded onto 96-well plates. The next day, cells were exposed to compounds at indicated concentrations and cultured at 37°C for 24 h. Next, U2OS cells were fixed in 4% PFA and nuclei were stained using Hoechst. Images were acquired at 4× magnification with ImageXpress Pico System (Molecular Devices, USA). Nuclei count was done using automatic segmentation using CellReporterXpress (Molecular Devices, USA). As AML cells grow in suspension, CellTiter-Glo Luminescent Assay (CTG, Promega, G7571) was used to assess viability. For clonogenic survival assays, U2OS cells were plated onto 12-well plates (7,000 cells/well). The following day, cells were exposed to compounds at the indicated concentrations. Every other day, the medium was replaced with fresh medium containing compounds until day 10. At the end of the experiment, cells were fixed and stained with 0.4% methylene blue in methanol for 30 min and images acquired with an image analyzer (Amersham Imager 600, GE Healthcare).

## Polysome profiling

MCF7 cells ($7 \times 10^6$ cells/plate) were seeded in 15-cm petri dishes. After 24 h, cells were treated as indicated for 20 h. Cells were washed and harvested in PBS containing CHX (100 μg/ml). After centrifugation, cells were resuspended in lysis buffer (5 mM Tris-HCl (pH 7.5), 2.5 mM MgCl2, 1.5 mM KCl, and 1x protease inhibitor cocktail (EDTA-free), CHX (100 μg/ml), DTT (2 μM), and 100 units of RNAse inhibitor) followed by addition of Triton X-100 (0.5%) and Sodium Deoxycholate (0.5%) (as indicated in [37]). Nuclei were removed by centrifugation (14,000 x $g$ for 10 min at 4°C). Then, lysates were loaded onto sucrose density gradients (15% to 50% in 20 mM HEPES (pH 7.6), 0.1 mM KCl, 5 mM MgCl$_2$, CHX (10 μg/ml), 0.1x protease inhibitor cocktail (EDTA-free), 10 units of RNAse inhibitor). After ultracentrifugation (3 h 10 min, 37,000 rpm at 4°C using a SW41Ti rotor), gradients were analyzed in a piston gradient fractionator (Biocomp). Profiles were acquired with Gradient profiler v.2.0 (Biocomp, Spain) and represented using Graphpad Prism.

## Transmission electron microscopy

For the TEM analyses, U2OS ($1 \times 10^6$ cells) were plated on 15 cm petri dishes. The following day, cells were fixed at RT in 2% glutaraldehyde in 0.1 M phosphate buffer (pH 7.4). After fixation, the cells were rinsed in 0.1 M phosphate buffer and centrifuged. Cell pellets were postfixed in 2% osmium tetroxide in 0.1 M phosphate buffer (pH 7.4) at 4°C for 2 h. Cells were then stepwise dehydrated in ethanol, followed by acetone and finally embedded in LX-112. Ultrathin sections (approximately 50 to 60 nm) were prepared using a Leica EM UC7 and contrasted with uranyl acetate followed by lead citrate. TEM imaging was done in a Tecnai 12 Spirit Bio TWIN transmission electron microscope operated at 80 kV and digital images acquired using a Veleta camera (Olympus Soft Imaging Solutions, Germany).

## Mass spectrometry

For the MS experiments, U2OS cells ($1 \times 10^6$) were seeded in 10 cm petri dishes. The next day, cells were treated with SKI-II (10 μM) or DMSO for 6 h. Each treatment was done in triplicate. Samples were digested with trypsin using S-trap devices. Peptides were subsequently labeled with TMT-11plex reagents and prefractionated by means of high pH-reverse phase

chromatography. Fractions were finally analyzed by LC–MS/MS analysis using a Q Exactive Plus (Thermo Fisher Scientific) coupled to an Ultimate 3000 nanoHPLC system. Raw files were analyzed with MaxQuant against a human protein database. Differential analysis was done with ProStar (v1.14) [51]: Proteins with $p$-value <0.05 and log2FC >0.3 (<−0.3) were defined as regulated (FDR 5% to 10% was estimated by Benjamini–Hochberg). GSEA (v 4.0.2) was performed using the preRanked function (enrichment statistic = weighted). Log2 ratios from each of the comparisons were used to rank all the proteins detected. Gene sets were extracted from Gene Ontology using the Molecular Signature Database (MSigDB), and pathways with an FDR q value < 0.05 were considered as statistically enriched.

## Statistics

Statistical parameters and tests are reported in the figures and corresponding figure legends. Statistical analysis was done using GraphPad Prism version 7.0 (GraphPad Software). One-way-ANOVA was performed for all the datasets that required comparison among multiple data points within a given experimental condition.

## Supporting information

**S1 Fig. Characterization of hit compounds stimulating OPP incorporation.** (A) Protein synthesis rates (a.u.) of 7 drugs identified as potential up-regulator hits in the primary screen after exposing U2OS to different concentrations (1, 3, and 5 μM) of the compounds for 24 h. Hit validation was conducted using both OPP (ORANGE) and HPG (BLUE) labeling assays. Data A in S5 Data. (B) HTM-mediated quantification of OPP signal in U2OS cells exposed for 3 h to 3 concentrations (1, 3, and 10 μM) of the compounds annotated as up-regulators of OPP incorporation. Torin 2 (0.5 μM) and CHX (100 μg/mL) were included as controls. DMSO control appears as a GRAY line, and a range of 15% appears as a dashed GRAY line. Data B in S5 Data. (C) HTM-mediated quantification of OPP signal in U2OS (up), MCF7 (medium), and RPE (bottom) cells in the presence of insulin (5 nM and 0.1 μM, 4 h). Compound treatments were done in cells growing in complete media (10% FCS) or in cells starved overnight in media with no FCS (starvation). Every dot represents the mean value of the measurements taken by well from 3 independent experiments. Data C in S5 Data. Statistical analyses were done using one-way ANOVA tests. ***$p$ < 0.001. A WB showing the phosphorylation status of the mTORC1 targets P70S6K and 4E-BP1 for each condition is shown below every panel. Vinculin was used as loading control. CHX, cycloheximide; DMSO, dimethyl sulfoxide; FCS, fetal calf serum; HPG, homopropargylglycine; HTM, high-throughput microscopy; OPP, o-propargyl-puromycin.
(EPS)

**S2 Fig. SKI-II reduces translation through activation of the ISR and independently of mTOR.** (A) HTM-mediated quantification of OPP incorporation in U2OS cells exposed to DMSO, SKI-II (10 μM), Torin 2 (0.5 μM), or SKI-II and Torin 2 for 3 h. Data A in S6 Data. (B) HTM-mediated quantification of OPP incorporation in MCF7 cells exposed to SKI-II (10 μM), SKI-II with PERKi (0.5 μM), PERKi (0.5 μM), or DMSO for 20 h. Data B in S6 Data. (C) Polysome profiles of MCF7 cells treated as in (B). Torin 2 (0.1 μM) was included as an additional control. Data C in S6 Data. DMSO, dimethyl sulfoxide; HTM, high-throughput microscopy; ISR, integrated stress response; mTOR, mechanistic target of rapamycin; OPP, o-propargyl-puromycin; PERKi, PERK inhibitor.
(EPS)

**S3 Fig. Comparison of the effects of SKI-II to the ISR-activating compound Tunicamycin.**
(A–C) Quantification from a Cell Painting experiment in U2OS cells preexposed for 1 h to
PERKi (0.5 μM) or DMSO, and then exposed to SKI-II (10 μM) or DMSO for 3 h. ER (A),
Golgi (B), and mitochondria (C) were labeled with Concanavalin A, MitoTracker, and wheat
germ agglutinin, respectively, and signals were quantified by HTM. Data A–C in S7 Data. (D)
Representative TEM images of U2OS cells treated with Tunic. (10 μg/mL) for 3, 6, and 24 h.
Cells were exposed to DMSO as a control. White arrows point at the ER. Scale bars, 0.5 μm.
(E) HTM-mediated quantification of OPP signal in U2OS cells preexposed for 1 h to PERKi
(0.5 μM), ISRIB (50 nM), or DMSO, and then exposed to Tunic. (10 μg/mL) or DMSO for 3 h.
Treatment with SKI-II (10 μM) was included in the assay. Data E in S7 Data. $^{***}p < 0.001$. (F)
HTM quantification of ATF4 nuclear signal in U2OS cells treated as in (E). Data F in S7 Data.
(G, H) HTM-dependent quantification of nuclei counts (normalized to DMSO) of U2OS cells
exposed to Tunic. (1 and 10 μg/mL) alone or Tunic. together with PERKi (0.5 μM) (G) or
ISRIB (10 nM) (H) for 24 h. Data G and H in S7 Data. $^{***}p < 0.001$. ATF4, activating tran-
scription factor 4; DMSO, dimethyl sulfoxide; ER, endoplasmic reticulum; HTM, high-
throughput microscopy; ISR, integrated stress response; ISRIB, integrated stress response
inhibitor; OPP, o-propargyl-puromycin; PERKi, PERK inhibitor; TEM, transmission electron
microscopy; Tunic., Tunicamycin.
(EPS)

**S4 Fig. ABC294640 damages the ER, activates the ISR, and kills cells independently of
SPHKs.** (A) HTM quantification of OPP incorporation in U2OS exposed to SKI-II (10 μM) or
ABC294640 (50 μM) for 3 and 6 h. Data A in S8 Data. $^{***}p < 0.001$ (B) HTM-dependent quan-
tification of ATF4 mean nuclear intensity in U2OS preexposed for 1 h to PERKi (0.5 μM),
ISRIB (50 nM), or DMSO, and then exposed to ABC294640 (50 μM) for 3 h. Data B in S8
Data. $^{***}p < 0.001$. (C) WB of markers of activation of the ISR (ATF4 and p-eIF2α) of parental
U2OS cells (control) and in U2OS cells deficient for SPHK1 (SPHK1 KO), SPHK2 (SPHK2
KO), or both (SPHK1&2 KO), as shown with SPHK1 and SPHK2 antibodies, in the presence
of ABC294640 (50 μM, 3 h) for 3 h. (D) HTM-mediated quantification of OPP incorporation
in parental U2OS cells (control) and in U2OS cells deficient for SPHK1 (SPHK1 KO), SPHK2
(SPHK2 KO), or both (SPHK1&2 KO) exposed to ABC294640 (50 μM, 6 h). Data D in S8
Data. Every dot represents the mean value of the measurements taken by well from 3 indepen-
dent experiments. Statistical analyses were done using one-way ANOVA tests. $^{***}p < 0.001$.
(E) HTM-dependent quantification of nuclei numbers (normalized to each DMSO) in paren-
tal U2OS cells (control) and in U2OS cells deficient for SPHK1 (SPHK1 KO), SPHK2 (SPHK2
KO), or both (SPHK1&2 KO) exposed to ABC294640 (50 μM) for 24 h. Data E in S8 Data. Sta-
tistical analyses were done using one-way ANOVA tests. $^{***}p < 0.001$. ATF4, activating tran-
scription factor 4; DMSO, dimethyl sulfoxide; ER, endoplasmic reticulum; HTM, high-
throughput microscopy; ISR, integrated stress response; ISRIB, integrated stress response
inhibitor; OPP, o-propargyl-puromycin; p-eIF2α, phosphorylated eIF2α; PERKi, PERK inhib-
itor; SPHK, sphingosine kinase; WB, western blotting.
(EPS)

**S1 Table. Results of the primary chemical screen.**
(XLS)

**S2 Table. Hits selected for validation from the primary screen.**
(XLS)

**S3 Table. Results of the validation chemical screen.**
(XLSX)

**S1 Data. Numerical data for Fig 1.**
(XLSX)

**S2 Data. Numerical data for Fig 2.**
(XLSX)

**S3 Data. Numerical data for Fig 3.**
(XLSX)

**S4 Data. Numerical data for Fig 4.**
(XLSX)

**S5 Data. Numerical data for S1 Fig.**
(XLSX)

**S6 Data. Numerical data for S2 Fig.**
(XLSX)

**S7 Data. Numerical data for S3 Fig.**
(XLSX)

**S8 Data. Numerical data for S4 Fig.**
(XLSX)

**S1 Raw images. Original immunoblots from Figs 2C, 2E, 4E, S1C, S4C.**
(PDF)

## Acknowledgments

The authors want to thank the Laboratories for Chemical Biology at Karolinska Institutet (LCBKI) for their help with the chemical screening, the Electron Microscopy (EMil) Unit at Karolinska Institutet for TEM analyses, Ola Larsson for help with polysome profiling, Javier Muñoz and Eduardo Zarzuela for help with proteomics, Kirsten Tschapalda for help in the screening analysis, and Jaime Espinoza, Martin Haraldsson, Andrea Björkman, and Emilio Lecona for valuable discussions throughout the project.

## Author Contributions

**Conceptualization:** Oscar Fernandez-Capetillo.

**Formal analysis:** Alba Corman, Jiri Bartek, Jordi Carreras-Puigvert.

**Funding acquisition:** Oscar Fernandez-Capetillo.

**Investigation:** Alba Corman, Dimitris C. Kanellis, Patrycja Michalska, Maria Häggblad, Vanesa Lafarga.

**Methodology:** Alba Corman, Dimitris C. Kanellis.

**Project administration:** Jordi Carreras-Puigvert, Oscar Fernandez-Capetillo.

**Supervision:** Jordi Carreras-Puigvert, Oscar Fernandez-Capetillo.

**Visualization:** Jordi Carreras-Puigvert.

**Writing – original draft:** Jordi Carreras-Puigvert, Oscar Fernandez-Capetillo.

**Writing – review & editing:** Alba Corman, Jiri Bartek, Jordi Carreras-Puigvert, Oscar Fernandez-Capetillo.

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
