## [Editor Report · Decision Letter 0]

10 Aug 2020

Dear Dr Fernandez-Capetillo, 

Thank you for submitting your manuscript entitled "A chemical screen identifies a link between lipid metabolism and mRNA translation" for consideration as a Research Article by PLOS Biology.

Your manuscript has now been evaluated by the PLOS Biology editorial staff as well as by an academic editor with relevant expertise and I am writing to let you know that we would like to send your submission out for external peer review.

Please re-submit your manuscript within two working days, i.e. by Aug 12 2020 11:59PM.

Kind regards,

Aaron

Aaron Nicholas Bruns, Ph.D.,

Associate Editor

PLOS Biology

---

## [Decision Letter · Decision Letter 1]

23 Sep 2020

Dear Dr Fernandez-Capetillo,

Thank you very much for submitting your manuscript "A chemical screen identifies a link between lipid metabolism and mRNA translation" for consideration as a Research Article at PLOS Biology. Thank you also for your patience as we completed our editorial process, and please accept my apologies for the delay in providing you with our decision. Your manuscript has been evaluated by the PLOS Biology editors, an Academic Editor with relevant expertise, and by three independent reviewers.

The reviews of your manuscript are appended below. You will see that the reviewers find your results potentially interesting for the RNA field, however they also raise several serious concerns that we would need to see addressed before considering the manuscript further for publication. Among the issues, the reviewers request that you explain in more detail why only cytoplasmic signals were in your OPP experiments. They also ask for further discussion of the lack of drugs that increased translation in the screen and that you provide further validation of this result either through using more physiological conditions as suggested by Reviewer 1 or through other methods.

Based on their specific comments and following discussion with the academic editor, I regret that we cannot accept the current version of the manuscript for publication. We remain interested in your study and we would be willing to consider resubmission of a much-revised version that thoroughly addresses all the reviewers' comments. We cannot make any decision about publication until we have seen the revised manuscript and your response to the reviewers' comments. Your revised manuscript would be sent for further evaluation by the reviewers.

We appreciate that these requests represent a great deal of extra work, and we are willing to relax our standard revision time to allow you six months to revise your manuscript.We expect to receive your revised manuscript within 6 months.

**IMPORTANT - SUBMITTING YOUR REVISION**

*Resubmission Checklist*

*Published Peer Review*

*PLOS Data Policy*

*Blot and Gel Data Policy*

Sincerely,

Aaron

Aaron Nicholas Bruns, Ph.D.,

Associate Editor,

abruns@plos.org,

PLOS Biology

Reviewers' comments:

Reviewer #1: The paper by Corman et al explores the effects of drugs on protein synthesis. The authors generated a high through put chemical screen to identify possible regulators of protein synthesis. Up regulators and down regulators. Based on this work two major discoveries were made: 1. Drugs do not promote protein synthesis above the level found under regular tissue culture conditions. 2. In addition to mTOR inhibitors, sphingosine kinase inhibitors downregulate protein synthesis in a mechanism that involves the induction of ER stress and eIF2a phosphorylation. This can be reversed by PERK inhibitors and the integrated stress response inhibitor ISRIB.

While this broad pharmacological analysis may be of general interest, I find this publication suitable better to a niche journal in pharmacology, such as JPET. This conclusion is based on the following general comments:

1. The initial screen identified 54 drugs that potentially induce protein synthesis in U2OS cells. A second validation excluded all. This put a big question mark on the validity of the screen, as the authors acknowledge. The authors should look for conditions that avoid these false positive outputs.

2. The authors use a concentration of 10uM of the SK inhibitors. While this is not a tremendously high concentration, it is considerably high for kinase inhibitors. Dose responses and relevance to the clinical concentrations should be done. These data should be accompanied by analysis of sphingosine phosphorylation to determine if this is an off target effect of the drug.

Many drugs induce ER stress at supra-pharmacological concentrations (celecoxib for example), a feature that is not related to the drugs primary pharmacological activity. This needs to be validated. Indications that sphingosine kinase inhibitors are inducing ER stress were already documented, so this is not a novel finding.

3. The fact that no drugs induced protein synthesis is important. However, this must be put in a more physiological context. For example, DMEM contains ample of amino acids, while serum does not. It is plausible that under ample of AA, glucose and serum supplementation the cells maximize protein synthesis. The authors should think about a relevant condition in which drugs may downregulate this and then repeat the screen under more stringent conditions. This may increase the impact of the study.

Minor issues:

The authors should quantify the expression and activity of SPHK 1 and 2 in their cells. This is important to conclude whether the two kinases are affecting similarly the stress responses.

The ATF4 antibody has background which may affect the immunofluorescence credibility. Please include a validation of the staining.

The fact that newly synthesized proteins are found in the nucleus is not surprising after 3 or 24 hours of labeling. Why the nuclear proteins were excluded from the analysis? Do the authors see cytoplasmic proteins only under shorter pulse conditions? Why the conditions of the screen were determined as done? This deserves explanation.

The authors should quantify total ceramide levels in the cells when ER toxicity is observed. Is it due to impairment in ceramide levels (very low in the ER) or is it something else? Is it related to the signaling of S1P? This may add more mechanistic understanding to the phenomenon.

Reviewer #2: Translation is essential to support cell survival and proliferation and defects in the control of this cellular process were linked to the development of several diseases. Translation requires significant amount of energy and this biological process is notoriously known to be regulated by the presence of growth factors and nutrients. Over the years, several groups have shown that the PI3K/mTOR pathway, a signaling node that senses nutrients and growth factors, plays a central role in regulating translation. Whether translation can be regulated by other pathways is still elusive. Here, Corman et al performed a chemical screen to identify new pharmacological regulators of translation. This screen, performed using more than 4000 compounds, confirmed that PI3K/mTOR inhibition strongly represses translation in vitro. None of the compound tested consistently increased translation. Here, the authors report that sphingosine kinase (SPHK) inhibitors significantly reduced translation in cells. The authors show that SHPK inhibitors repress translation by altering ER membranes integrity and by activating the UPR. Overall, this manuscript is well written and the results presented are clear. The experimental approaches are well described and appropriate. The conclusions drawn by the authors are, most of the time, supported by the data presented. The major concern I have regarding this work relates to the novelty of the results presented. The most important regulators of translation identified from the screen are part of the PI3K/mTOR pathway, a signalling node that was already established as a key player in regulating translation. The other molecules that showed an effect on translation were inhibitors of SHPK, a set of molecules that were previously shown to activate the UPR. Considering the well-established roles of the UPR on the repression of translation, it is not clear to me how the current manuscript provides real conceptual advances in the understanding of the cellular processes that control translation. Defining the precise impact of SPHK inhibitors on the lipid composition of the ER membranes and trying to determine how these changes translates into the activation of the UPR could represent a great addition to this paper.

1) The authors mention that a part of the OPP signal is located to the nucleus and then conclude that 'as translation takes place in the cytoplasm, we used the quantification of cytoplasmic OPP signal for subsequent analysis'. Because OPP labelled proteins can theoretically diffuse in the cell to reach other cellular compartments, it is not clear to me why only the cytosolic amount of OPP signal was quantified. More explanations should be provided.

2) Have the authors measured the linearity of the OPP staining before running the screen? This should be discussed. 

3) It is not clear to me how 54 compounds, identified as up-regulators of translation in the primary screen, all failed in the replication screen. This indicates that either i) the criteria to include these molecules as hits in the primary screen were too inclusive or ii) that a problem occurred in one the screen performed (primary or validation). The authors should comment.

4) The primary screen performed here to identify new regulators of translation was based on the use of 4,166 molecules. The most important regulators of translation identified from this screen were part of the PI3K/mTOR pathway, a signaling node that was already established as a key player in regulating translation. Overall these results were expected based on previous literature. The other molecules that showed an effect on translation were inhibitors of sphingosine kinase, a set of molecules that were previously shown to activate the UPR. Considering the well-established roles of the UPR on the repression of translation, it is not clear to me how the current manuscript provides real conceptual advances in the understanding of the cellular processes that control translation. 

5) Defining the precise impact of SPHK inhibitors on the lipid composition of the ER membranes and trying to determine how these changes translates into the activation of the UPR could represent a great addition to this paper.

6) Are the effects of Torin2 and SKI-II on translation additive? If SKI-II controls translation by activating the UPR, independently of mTOR, we could expect to see an additive effect of both compounds.

7) It would be interesting to probe for GRP78 levels in the experiment described in 2C and 2E. 

8) The phospho-PERK blot presented in Figure 2E is difficult to interpret. Another WB with an anti-phospho PERK antibody should be presented. Several antibodies against pPERK are commercially available.

9) After a careful review of the manuscript, I really think the authors should revise the title of their manuscript to better describe their findings and to be more specific. Here, the authors used a chemical screen and found that sphingosine kinase inhibition promotes UPR activation and represses translation. The current title of the manuscript 'A chemical screen identifies a link between metabolism and mRNA translation' is somehow misleading and does not summarize the data presented.

10) In the abstract and in the discussion, the authors mention 'On the other hand, and despite the large number of molecules tested, our study failed to identify chemicals substantially increasing translation, raising doubts on to what extent translation can be supra-physiologically stimulated in mammalian cells'. I do not think this conclusion is correct. The incapacity to promote translation to a very high level is likely specific to the experimental model used. Here, all the experiments were performed in proliferating cancer cells grown in rich conditions (high energy, full serum). These cells likely maximize basal translation to support proliferation. Because the authors did not measure translation in normal mammalian tissues and in response to conditions that may impact on translation (ex. cell differentiation, cell proliferation, cell activation), it is incorrect to say that translation cannot be highly induced in mammalian cells. 

Minor

1) The authors should produce a table clearly listing the 48 compounds that decreased and the 54 compounds that increased translation. 

Reviewer #3: In the current manuscript, Corman et al. developed a chemical screening approach to identify small molecule modulators of general translation in a human cancer cell line (U2-OS). The authors tested a library of more than 4,000 compounds, but could not identify translational activators. However, they were able to recovered inhibitors of the mTOR/PI3K pathway as potent translational repressors. In addition, the authors identified sphingosine kinases, especially SPHK2, as molecular targets to block translation. The latter effect seems to depend on the induction of ER stress and the integrated stress response by the sphingosine kinase inhibitors. Consequently, the effects could be reversed by the simultaneous inhibition of both sphingosine kinase and PERK.

Overall, the manuscript is well-written and easy to follow and the results are of general interest to the RNA community. Nevertheless, I have some concerns that the authors should try to address:

1) Please elaborate a bit more on the selection and composition of the chemical library. Why were these inhibitors chosen? How many kinase inhibitors vs. epigenetic inhibitors vs. others, etc.?

2) Please improve the workflow scheme in Figure 1C. The pulse labelling with OPP for 1 h is not obvious and also the second incubation phase is 24 h, after 23 h CHX was added. Is this information really needed here? Also, it has to be explained why the nuclear counterstaining was done. It is just briefly indicated with (viability) in the text. Last but not least, the choice of the cell line system should be explained. I guess U2-OS were selected due to their rather large cytoplasmic compartment, correct? 

However, please see my next point.

3) Only the cytoplasmic OPP signal was considered for the screening analysis. However, proteins move to their target destination after synthesis. Hence, the whole-cell signal should be considered. How would this differ from the current analysis?

4) Please improve some of the Western blot images to allow more band width. For example, in Figure 2E the signals for PERK are nearly at the border / edge of the box. Also, a black line is visible. In line with this, Figure S1C should also show a larger section of the blot.

5) The damage of SKI-II to the ER membranes happens quite fast (after 3-6 h). However, it is unclear what really happens to them. The authors did not measure any lipids. Do these change?

6) How specific is the SKI-II or ABC294640 inhibitor? Is the effect truly mediated by SPHK2 inhibition? How effective is the inhibition, i.e. can the authors measure remaining activity? To analyze specificity one usually leverages genetic (knockout) models. The inhibitor should not cause any effects in such a system. Also, the authors should perform SPHK2 depletion experiments and analyze the effects on translation and stress pathways. 

7) Please discuss the obvious lack of ABC294640 activity in your screen and its clinical relevance given that effects were only achieved with a much higher (50 µM) dose.

---

## [Decision Letter · Decision Letter 2]

16 Apr 2021

Dear Dr Fernandez-Capetillo,

Thank you for submitting your revised Research Article entitled "A chemical screen for modulators of mRNA translation identifies a novel mechanism of toxicity for sphingosine kinase inhibitors" for publication in PLOS Biology. I'm handling your manuscript temporarily while my colleague Dr Ines Alvarez-Garcia is out of the office. We've now obtained advice from the original reviewers and have discussed their comments with the Academic Editor. 

You'll see that while reviewer #1 still harbours some reservations about the strength of your manuscript and its suitability for our journal, the other two reviewers are more positive, and after some discussion with the Academic Editor, we have decided that we will probably accept this manuscript for publication, provided you satisfactorily address the remaining points raised by the reviewers. Please also make sure to address the following data and other policy-related requests.

IMPORTANT:

a) Please address the remaining concerns raised by reviewer #2 (we note that may entail one or two very minor experiments and some textual changes, including to the title). It may also be helpful to address reviewer #1's points textually, as these objections may also occur to future readers.

b) Please address our Data Policy requests further down. Specifically, you need to provide the underlying numerical values for Figs 1CEF, 2AFG, 3ABE, 4ABCDFGH, S1ABC, S2ABC, S3ABCEFGH, S4ABDE and cite the location of the data in each relevant Figure legend.

We expect to receive your revised manuscript within two weeks. 

*Published Peer Review History*

*Early Version*

Sincerely,

Roli Roberts

Roland Roberts PhD

Senior Editor

PLOS Biology

on behalf of

Ines Alvarez-Garcia, PhD,

Senior Editor,

ialvarez-garcia@plos.org,

PLOS Biology

DATA POLICY:

Regardless of the method selected, please ensure that you provide the individual numerical values that underlie the summary data displayed in the following figure panels as they are essential for readers to assess your analysis and to reproduce it: Figs 1CEF, 2AFG, 3ABE, 4ABCDFGH, S1ABC, S2ABC, S3ABCEFGH, S4ABDE. NOTE: the numerical data provided should include all replicates AND the way in which the plotted mean and errors were derived (it should not present only the mean/average values).

DATA NOT SHOWN?

REVIEWERS' COMMENTS:

Reviewer #1:

I don't have any further scientific comments.

The fact that drugs have off target effects at high concentrations is not unusual. The authors indicate the development of a chronic ER stress condition that merits caution.

I followed the clinical trial data and it seems that ABC294640 at 500 mg per day is tolerated well. If this molecule would cause chronic ER stress conditions, I assume that blood glucose levels would have risen. No reports were documented.

I think that the lack of a mechanism by which SKII inhibitors is working off-target detracts my enthusiasm. My opinion is that PLOS Biology is seems to be above the scientific significance of these findings.

Reviewer #2:

I want to thank the authors for this new version of the manuscript. They provided answers for most of my comments. Here are minor comments that should be addressed.

Comments on the revised manuscript.

1) The authors should review the abstract to remove this sentence : 'None of the compounds upregulated translation, which could be due to the screen being performed in cancer cells grown in full media where translation is already present at very high levels'. This information is not useful in the abstract. 

2) Line 6 - change mammalian for mechanistic

3) Loading control S4C is weird (and the levels of SPHK2 are highly variable). The authors should provide explanation or run the experiment again.

4) Uniformize the use of SK-II and SKI-II thought the manuscript.

5) Figure 2C - why does Torin 2 reduces total S6K levels? Are the authors sure that the right antibody was used? Looks like pS6K is presented instead of total S6K.

Comment on the responses provided.

Comment 9 : The title suggested is better but still not accurate. This title 'A chemical screen for modulators of mRNA translation identifies a novel mechanism of toxicity for sphingosine kinase inhibitors' is incorrect considering that the mechanism of toxicity is not identified/provided. The data presented indicate that SPHK inhibitors blocks translation independently of SPHK1/2, but the mechanism is still unclear.

Reviewer #3:

The authors have fully answered my previous concerns and I have no further questions. However, they need to include the sgRNA sequences used to generate SPHK1/2 CRISPR KOs.

---

## [Editor Report · Decision Letter 3]

5 May 2021

Dear Dr Fernandez-Capetillo,

On behalf of my colleagues and the Academic Editor, Wendy Gilbert, I am pleased to say that we can in principle offer to publish your Research Article entitled "A chemical screen for modulators of mRNA translation identifies a distinct mechanism of toxicity for sphingosine kinase inhibitors" in PLOS Biology, provided you address any remaining formatting and reporting issues. These will be detailed in an email that will follow this letter and that you will usually receive within 2-3 business days, during which time no action is required from you. Please note that we will not be able to formally accept your manuscript and schedule it for publication until you have made the required changes.

PRESS

Thank you again for supporting Open Access publishing. We look forward to publishing your paper in PLOS Biology. 

Sincerely, 

Ines

--

Ines Alvarez-Garcia, PhD 

Senior Editor 

PLOS Biology